# KD-BIRL: Kernel Density Bayesian Inverse Reinforcement Learning

**Aishwarya Mandyam**                                                                                 *am2@stanford.edu*
*Department of Computer Science*
*Stanford University*

**Didong Li**                                                                                         *didongli@unc.edu*
*Department of Biostatistics*
*University of North Carolina*

**Diana Cai**                                                                                *dcai@flatironinstitute.org*
*Flatiron Institute*

**Andrew Jones**                                                                                    *aj13@princeton.edu*
*Department of Computer Science*
*Princeton University*

**Barbara E. Engelhardt**                                                                       *barbarae@stanford.edu*
*Gladstone Institutes*
*Department of Biomedical Data Science*
*Stanford University*

**Reviewed on OpenReview:** *https://openreview.net/forum?id=B8OWUNhTAw*

## Abstract

Inverse reinforcement learning (IRL) methods infer an agent's reward function using demonstrations of expert behavior. A Bayesian IRL approach models a distribution over candidate reward functions, capturing a degree of uncertainty in the inferred reward function. This is critical in some applications, such as those involving clinical data. Typically, Bayesian IRL algorithms require large demonstration datasets, which may not be available in practice. In this work, we incorporate existing domain-specific data to achieve better posterior concentration rates. We study a common setting in clinical and biological applications where we have access to expert demonstrations and known reward functions for a set of training tasks. Our aim is to learn the reward function of a new test task given limited expert demonstrations. Existing Bayesian IRL methods impose restrictions on the form of input data, thus limiting the incorporation of training task data. To better leverage information from training tasks, we introduce kernel density Bayesian inverse reinforcement learning (KD-BIRL). Our approach employs a conditional kernel density estimator, which uses the known reward functions of the training tasks to improve the likelihood estimation across a range of reward functions and demonstration samples. Our empirical results highlight KD-BIRL's faster concentration rate in comparison to baselines, particularly in low test task expert demonstration data regimes. Additionally, we are the first to provide theoretical guarantees of posterior concentration for a Bayesian IRL algorithm. Taken together, this work introduces a principled and theoretically grounded framework that enables Bayesian IRL to be applied across a variety of domains.

# 1 Introduction

Inverse reinforcement learning (IRL) is a strategy within the domain of reinforcement learning (RL) that infers an agent's reward function from demonstrations of the agent's behavior. The learned reward function helps identify the objectives driving the agent's behavior. IRL methods have been successfully applied across several domains, including robotics (Ratliff et al., 2006a; Kolter et al., 2008), swarm animal movement (Ashwood et al., 2020), and ICU treatment (Yu et al., 2019a). The earliest IRL algorithms learned point estimates of the reward function (Ng & Russell, 2000; Abbeel & Ng, 2004), and were applied to problems in path planning (Mombaur et al., 2010) and urban navigation (Ziebart et al., 2009).

Despite the success of early IRL approaches, there are limitations to inferring a point estimate of the reward function. First, the reward function is frequently non-identifiable (Abbeel & Ng, 2004; Ratliff et al., 2006b; Ziebart et al., 2009), meaning there are multiple reward functions that explain the expert demonstrations equally well. Second, for finite demonstration data, the point estimate of the reward function fails to capture the noise in the data-generating process, and does not model uncertainty in the resulting reward function estimate. A Bayesian approach learns a posterior distribution over reward functions after observing all samples. This distribution places mass on reward functions proportional to how well they explain the observed behavior, using the information from the samples (Ramachandran & Amir, 2007; Michini & How, 2012b; Choi & Kim, 2012; Balakrishnan et al., 2020; Chan & van der Schaar, 2021; Michini & How, 2012a), better highlights possible alternative explanations of agent objectives.

Despite their ability to produce a set of candidate reward functions, Bayesian IRL algorithms typically require large amounts of high-quality expert demonstration data to ensure posterior contraction (Brown et al., 2020). A Bayesian posterior changes depending on the samples that are observed. "Contraction" is the phenomenon of the posterior concentrating at the true value as more samples are observed. However, in some applications, such as those involving clinical or biological data, expert demonstrations are expensive to obtain. For example, in a clinical application, collecting expert demonstrations may involve parsing electronic health records (EHRs) (Johnson et al., 2016), which contain information associated with patient trajectories as well as clinical notes (Johnson et al., 2023). However, each clinician treats a limited number of patients, so there are few patient trajectories. In a biological application, we might be interested in studying the factors that drive the hunting behavior of T-cells, which affects how fast they kill tumor cells. For example, certain conditions may induce "pack hunting" in which a T-cell works with other T-cells to kill tumor cells. One common approach to gather information about how cells move under different conditions is genetic perturbation studies (Feldman et al., 2022), which examine the cellular behavior changes following an induced genetic perturbation. Previous work has studied how some genetic perturbations, such as RASA2 (Carnevale et al., 2022) and CUL5 knockout Liao et al. (2024), affect the efficiency of T cells killing tumor cells. However, each study is expensive to run, and thus, a limited number of trajectories are available. In this work, we use supplementary data sources to improve the posterior contraction rate for Bayesian IRL algorithms in applications with few expert demonstration trajectories.

We assume that we have several training tasks and a single test task. We have access to expert demonstrations as well as the associated reward function for each training task. This assumption is reasonable in many application areas. Referring to the examples above, in clinical applications, the reward function represents the clinical treatment objective. These treatment objectives can be obtained from EHR clinical notes, which contain clinician-transcribed goals of treatment. In the biophysical movement example, the reward function represents how various factors, such as the density of the tumor cells and the proliferation rate of T-cells, induce a certain type of hunting behavior. In both examples, each training task is associated with a known reward function. Our goal is to infer the reward function for the test task, i.e., a patient trajectories treated by a new clinician or cell behavior trajectories from a new genetic perturbation, with a small set of expert demonstrations.

Integrating supplementary data poses challenges within the current landscape of Bayesian IRL algorithms. Existing methods for single-task IRL (Ramachandran & Amir, 2007; Michini & How, 2012b) are restricted to a framework where the set of expert demonstrations arises from a single task. Alternative formulations, such as meta-IRL methods, can accommodate demonstrations from several distinct tasks. However, meta-IRL methods cannot incorporate information about the known reward functions from the training tasks.

In both the single-task and meta-IRL settings, one can learn an informative prior to facilitate the reward function inference. A "prior" is the parameterized Bayesian distribution before observing samples. Prior distributions can be uninformative (i.e., uniform), or informative (i.e., Gaussian). Nevertheless, earlier work has shown that using an informative prior may introduce bias (Gelman et al., 2020) or reduce sensitivity to new demonstrations from the test task (Kass & Wasserman, 1996). Ultimately, neither the single-task nor meta-IRL approaches is suitable because the methods either cannot incorporate demonstrations from training tasks, or cannot adequately include information about the known reward functions from training tasks.

To incorporate all possible information sources stored within the training tasks, we introduce kernel density Bayesian inverse reinforcement learning (KD-BIRL). In our work, we use conditional kernel density estimation (CKDE) to approximate the likelihood function. The "likelihood" is a probability distribution that defines how likely a sample is to be observed, given parameter values for a specific probabilistic model. Our CKDE formulation uses kernels to quantify how much each demonstration sample contributes to the likelihood, conditioned on a reward function. The use of a CKDE allows us to incorporate reward functions from the training tasks into the likelihood estimation stage.

Our work demonstrates that a posterior distribution learned using a likelihood estimated by CKDE empirically contracts faster than single-task IRL approaches. This holds true even if the single-task approaches have informative priors. The contributions of our work follow.

1. We propose KD-BIRL, which incorporates supplementary data by using conditional kernel density estimation to approximate the likelihood (Section 4). To the best of our knowledge, we are the first to identify and formalize a setting in which multiple sets of demonstrations and their corresponding reward functions exist as training tasks, and the first to build a Bayesian IRL algorithm that can take advantage of it.

2. We perform asymptotic analysis, and demonstrate that the posterior distribution contracts to the equivalence class of the true reward function (Section 5). We are also the first to provide asymptotic guarantees of posterior contraction in a Bayesian IRL setting.

3. With a feature-based reward function (Section 4.2), KD-BIRL infers a posterior distribution over reward functions in high-dimensional state spaces, empirically contracting using fewer samples than baseline approaches (Figure 4).

We first review related work in Section 2 and fundamental topics in Section 3. Then, we formally introduce our problem and KD-BIRL in Section 4. We provide posterior consistency guarantees in Section 5. In Section 6, we demonstrate results in a Gridworld environment and a challenging sepsis management task, benchmarking KD-BIRL against existing Bayesian IRL methods. Finally, in Section 7, we discuss broader implications.

## 2  Related Work

**Single-task Bayesian IRL**. Single-task Bayesian IRL methods receive as input a set of demonstration data that originates from an expert policy optimizing for an unknown reward function. There have been many follow-up methods which improve upon the original Bayesian IRL algorithm (Ramachandran & Amir, 2007). To perform inference for more complex reward function structures, several approaches learn nonparametric reward functions, which use strategies such as Gaussian processes (Levine et al., 2011; Qiao & Beling, 2011) and Indian buffet process (IBP) priors (Choi & Kim, 2013). Šošić et al. (2018) decompose a complex global IRL task into simpler sub-tasks. Other approaches reduce computational complexity by either using informative priors (Rothkopf & Ballard, 2013), different sampling procedures (e.g., Metropolis-Hastings (Choi & Kim, 2012), expectation maximization (Zheng et al., 2014)), variational inference (Chan & van der Schaar, 2021)), or learning several reward functions that each describe a subset of the state space (Michini & How, 2012b;a). In contrast, we assume access to demonstration data from experts optimizing for multiple tasks, and use the known reward functions of these tasks to facilitate reward function inference for a new task.

**Meta-IRL and multi-task IRL**. Meta-IRL algorithms receive a set of unstructured demonstrations that arise from several distinct but unknown tasks, each associated with a different reward function. Previous work learns a context-conditional reward function from training tasks, which can then be generalized to unseen but similar tasks (Yu et al., 2019b; Xu et al., 2019). However, these approaches learn a point estimate of the reward function instead of inferring a posterior distribution. Additionally, these approaches do not incorporate information about the reward functions associated with each of the training tasks. Multi-task IRL algorithms (Dimitrakakis & Rothkopf, 2012; Gleave & Habryka, 2018; Arora et al., 2021; Chen et al., 2023; Choi & Kim, 2012) are similar, but focus on learning task-specific reward functions rather than generalizing to new tasks.

**IRL with supplementary information**. There are a few methods that use both expert demonstration data and supplementary data sources to solve IRL problems. One method finds that using a combination of sub-optimal demonstrations and high-quality human feedback can improve reward function inference (Ezzeddine et al., 2018). Another approach finds that expert demonstrations of varying quality aids in learning the reward function because the IRL model can learn from non-expert demonstrations (Audiffren et al., 2015). Brown et al. (2020) finds that Bayesian learning can be sped up using human preferences over demonstrations. Our work builds upon these efforts to improve Bayesian IRL using supplementary data sources. Specifically, we reformulate the likelihood function using CKDE to take advantage of a training dataset with both demonstrations and reward functions from multiple expert agents.

## 3 Preliminaries

### 3.1 Single-task Bayesian IRL

IRL methods infer an agent's reward function given expert demonstrations of its behavior. We can model the agent's behavior using a Markov decision process (MDP). The MDP is defined by $(\mathcal{S}, \mathcal{A}, P, \gamma)$, where $\mathcal{S}$ is the state space; $\mathcal{A}$ is a discrete set of actions; $P(\mathbf{s}_{t+1}|\mathbf{s}_t, a_t)$ defines state transition probabilities from time $t$ to $t+1$; and $\gamma \in [0, 1]$ is a constant discount factor.

The IRL algorithm receives as input $n$ expert demonstration samples, $\{(s_i^e, a_i^e)\}_{i=1}^n$, and each sample is a 2-tuple that specifies the *e*xpert agent's state and chosen action. The demonstration samples arise from an agent following the stationary policy $\pi^\star : \mathcal{S} \to \mathcal{A}$, which optimizes for a fixed but unknown ground truth reward function $R^\star : \mathcal{S} \times \mathcal{A} \to \mathbb{R}$, where $R^\star \in \mathcal{R}$ and $\mathcal{R}$ is the space of reward functions.

Given these expert demonstrations, Bayesian IRL algorithms treat $R$ as inherently random and aim to learn a posterior distribution over $R$ that can rationalize the expert demonstrations. By specifying a prior distribution over the reward space $\mathcal{R}$, $p(R)$, and a likelihood function for the observed data, $\prod_{i=1}^n p(s_i^e, a_i^e|R)$, Bayesian IRL methods infer a posterior distribution over $R$, using the expert demonstrations as evidence. By Bayes' rule, the posterior density of the reward function is given by:

$$p\left(R \,\middle|\, \{(s_i^e, a_i^e)\}_{i=1}^n\right) = \frac{p(R) \prod_{i=1}^n p(s_i^e, a_i^e|R)}{p(\{(s_i^e, a_i^e)\}_{i=1}^n)}. \tag{1}$$

The earliest formulation of Bayesian IRL used the optimal $Q$-value function to approximate the likelihood component, $p(s_i^e, a_i^e|R)$, of Equation 1 (Ramachandran & Amir, 2007). The likelihood takes the form

$$p(s, a \,|\, R) \propto e^{\alpha Q^\star(s, a, R)}, \tag{2}$$

where $\alpha > 0$ is an inverse temperature parameter characterizing the optimality of the expert demonstrations and $Q^\star(s, a, R)$ is the optimal $Q$-value function for reward function $R$, defined as

$$Q^*(s, a, R) = \max_\pi \mathbb{E}_\pi \left[ \sum_{t=0}^\infty R(S_t, A_t) \,\middle|\, S_0 = s, A_0 = a \right].$$

### 3.2 Problem Setting

In this work, we assume that we have already encountered several training tasks with different reward functions. For each training task, we have access to both optimal demonstrations from the corresponding

expert RL agent, and know the reward function the expert is optimizing for. Specifically, there are $m$ samples in the training dataset $\{(s_j, a_j, R_j)\}_{j=1}^m$, where each state-action tuple $(s_j, a_j)$ is a demonstration of an expert optimizing for the reward function $R_j$. We note that there will be several state-action pairs that correspond to the same reward function since there are several samples from each training task; therefore, $R_j$ is not unique. Our goal is to learn the unknown reward function $R^\star$ of a new test task given a limited amount of expert demonstrations of the new test task, $\{(s_i^e, a_i^e)\}_{i=1}^n, \ (n << m)$.

### 3.3 Conditional kernel density estimation

In this work, we propose approximating the likelihood in Equation 1 using conditional density estimation (CDE). In Section 4.1, we show that CDE allows us to incorporate information about both the reward functions and demonstrations associated with training tasks. Given a set of empirical observations, CDE aims to capture the relationship between the responsive variable $Y$ and the covariates $X$ by modeling the probability density function,

$$p(y|x) = \frac{p(x,y)}{p(x)}.$$

With this perspective, any appropriate conditional density estimator can be used to approximate the likelihood; examples include the conditional kernel density estimator (CKDE) (van der Vaart, 2000), Gaussian process models (Riihimäki & Vehtari, 2012), or diffusion models (Cai & Lee, 2023). In our work, we choose CKDE because it is nonparametric, which allows us to model complex reward structures. Additionally, CKDE has a closed form and is straightforward to implement (Holmes et al., 2007; Izbicki & Lee, 2016).

Specifically, the CKDE estimates the conditional density $p(y|x)$ by approximating the joint distribution $p(x,y)$ and marginal distribution $p(x)$ separately via kernel density estimation (KDE). Intuitively, KDE estimates the probability density function of a random variable by considering how close each observed data point is to a given location. The likelihood at that location is higher if there are more observations nearby, with closer observations contributing more to the estimate. Given pairs of vector observations $\{(x_j, y_j)\}_{j=1}^m$, the KDE approximations for the joint and marginal distributions are

$$\begin{aligned}
\widehat{p}_m(x,y) &= \frac{1}{m} \sum_{j=1}^m K\left(\frac{d_x(x,x_j)}{h}\right) K'\left(\frac{d_y(y,y_j)}{h'}\right), \\
\widehat{p}_m(x) &= \frac{1}{m} \sum_{j=1}^m K\left(\frac{d_x(x,x_j)}{h}\right),
\end{aligned} \tag{3}$$

where $K$ and $K'$ are kernel functions with bandwidths $h, h' > 0$ respectively, and $d_x, d_y$ are distance functions that measure the similarities between $x, x_j, y, y_j$, respectively. There are many choices for suitable kernels (e.g. uniform, normal, triangular) and distance metrics (e.g. Euclidean distance, cosine similarity, Manhattan distance) (Cawley & Talbot, 2010), which should be chosen depending on the application domain. To approximate the conditional density, the CKDE simply takes the ratio of these two KDE approximations:

$$\widehat{p}_m(y|x) = \frac{\widehat{p}(x,y)}{\widehat{p}(x)} = \sum_{j=1}^m \frac{K\left(\frac{d_x(x,x_j)}{h}\right) K'\left(\frac{d_y(y,y_j)}{h'}\right)}{\sum_{\ell=1}^m K\left(\frac{d_x(x,x_\ell)}{h}\right)}. \tag{4}$$

Note that the above approximation is not a distribution because it is not normalized.

## 4 Methods

There are several concerns with using Equation (2) as the likelihood function to learn the posterior distribution of the reward function of the test task. First, Equation (2) uses the optimal $Q$-value function, which is difficult to learn given a limited amount of demonstration data from the test task. As a result, a large demonstration dataset is required for posterior contraction. Furthermore, current literature often uses

parametric methods to estimate the optimal $Q$-function. As a result, if the function class is misspecified, the likelihood function is not guaranteed to converge to the true likelihood function.

To guarantee likelihood convergence, we use KDE, which is a non-parametric method, to learn a Bayesian IRL posterior. As mentioned earlier, we also use the information stored within the training task datasets to improve posterior contraction with limited test task demonstration data.

## 4.1 Kernel density Bayesian IRL

To avoid the issues with prior approaches, we propose kernel density Bayesian inverse reinforcement learning (KD-BIRL), which uses CKDE to approximate the likelihood. CKDE calculates the likelihood of a reward function sample by weighting the density estimate based on the distance to observations from the training data. The CKDE allows us to approximate the Bayesian IRL likelihood using the training dataset $\{(s_j, a_j, R_j)\}_{j=1}^m$ as

$$
\begin{aligned}
\widehat{p}_m(s, a \mid R) &= \frac{\widehat{p}_m(s, a, R)}{\widehat{p}_m(R)} \\
&= \sum_{j=1}^m \frac{K\left(\frac{d_s((s,a),(s_j,a_j))}{h}\right) K'\left(\frac{d_r(R,R_j)}{h'}\right)}{\sum_{\ell=1}^m K'\left(\frac{d_r(R,R_l)}{h'}\right)},
\end{aligned}
\tag{5}
$$

where $h, h' > 0$ are the bandwidth hyperparameters with larger values giving a smoother function estimate, and $d_s : (\mathcal{S} \times \mathcal{A}) \times (\mathcal{S} \times \mathcal{A}) \to \mathbb{R}_{\geq 0}$ and $d_r : \mathcal{R} \times \mathcal{R} \to \mathbb{R}_{\geq 0}$, specify the distance between state-action tuples and reward functions, respectively.

Given $n$ expert demonstration samples from the test task, $m$ training demonstration samples from the training tasks, and a prior $p(R)$, we now learn a posterior on $R$ as,

$$
\widehat{p}_m^n(R|\{s_i^e, a_i^e\}_{i=1}^n) \propto p(R) \prod_{i=1}^n \widehat{p}_m(s_i^e, a_i^e \mid R)
\tag{6}
$$

$$
= p(R) \prod_{i=1}^n \sum_{j=1}^m \frac{K\left(\frac{d_s((s_i^e,a_i^e),(s_j,a_j))}{h}\right) K'\left(\frac{d_r(R,R_j)}{h'}\right)}{\sum_{\ell=1}^m K'\left(\frac{d_r(R,R_l)}{h'}\right)}.
\tag{7}
$$

There are many possible options for the kernel $K$, including uniform and triangular kernels. In KD-BIRL, we use a Gaussian kernel because it can approximate bounded and continuous functions well. Several prior distributions can be appropriate for the prior distribution, $p(R)$ depending on the characteristics of the MDP, including Gaussian (Qiao & Beling, 2011), Beta (Ramachandran & Amir, 2007), or Chinese restaurant process (CRP) (Michini & How, 2012a). To define distance metrics on a vector space, one can choose cosine similarity, dot product similarity, Manhattan distance, etc. In our work, we choose *Euclidean distance* for $d_s, d_r$ and a uniform or Gaussian prior depending on the application domain. We choose the bandwidth hyperparameters $h, h'$ using rule-of-thumb procedures (Silverman, 1986). These procedures define the optimal bandwidth hyperparameters as the variance of the pairwise distance between the training data demonstrations and the training data reward functions respectively. Intuitively, one wants to choose the smallest bandwidth hyperparameter the data will allow, to avoid density estimates that are too smooth.

There are many techniques to sample from the posterior distribution (Equation 6), such as MCMC sampling Van Ravenzwaaij et al. (2018) and variational inference Blei et al. (2017). We use a Hamiltonian Monte Carlo algorithm (Team, 2011) (details in Appendix F, and Algorithm 1) which is suited to large parameter spaces. However, note that KD-BIRL introduces a new likelihood estimation approach, and that any posterior sampling algorithm (i.e. Markov Chain Monte Carlo (MCMC), Metropolis Hastings, Gibbs Sampling etc.) can be used. A key computational gain of our approach over the original Bayesian IRL algorithm (also a sampling-based approach) is that we avoid the cost of re-estimating the $Q$-value function with every iteration of sampling. This is possible because our posterior distribution (Equation 6) does not depend on $Q^\star$. Thus, sampling from the posterior is less computationally intensive than sampling from the standard Bayesian IRL algorithm posterior.

### 4.2 Feature-based reward function

While KD-BIRL can be applied directly in environments where the reward function has few parameters, kernel density estimation is known to scale poorly to high dimensions. In practice, the computational cost of the CKDE increases substantially as the number of reward function parameters and the number of behavior samples increase (Izbicki & Lee, 2017). To allow CKDE to perform reward inference in environments with higher-dimensional reward functions, we choose to re-parameterize the reward function.

We adopt a *feature-based reward function*, which allows us to represent the reward as a linear combination of a weight vector and a low-dimensional feature encoding of the state (Abbeel & Ng, 2004; Hadfield-Menell et al., 2017). Earlier work demonstrates the success of feature-based reward functions in imitation learning in complex control problems (Ratliff et al., 2006b; Ziebart et al., 2008; 2009; Brown et al., 2020). The feature-based reward function is written as $R(s,a) = w^\top \phi(s,a)$, where $w \in \mathbb{R}^q$ and $\phi : S \times A \to \mathbb{R}^q$. Here, $\phi$ is a known function that maps a state-action tuple to a feature vector of length $q$. This parameterization is advantageous because the dimension of the weights does not scale with the dimension of the state representation or the cardinality of the state space, and does not rely on the state space being discrete. Ideally, $q$ is large enough to fully represent the state-action space, but small enough for computational efficiency. Our work assumes that the featurized projection $\phi(s,a)$ fully captures the information in the given state-action tuple. The goal of a Bayesian IRL method is now to learn a posterior over $w$ rather than $R$.

Under the feature-based reward function paradigm, a sample in the training task dataset is now of the form $\{(s_j, a_j, w_j)\}_{j=1}^m$, where $w_j$ is the weight vector of length $q$ associated with the sample$(s_j, a_j)$. The posterior over the weights $w$ using the CKDE formulation is,

$$
\begin{aligned}
\widehat{p}_m(s, a \mid w) &= \frac{\widehat{p}_m(s, a, w)}{\widehat{p}_m(w)} \\
&= \sum_{j=1}^m \frac{K\left(\frac{d_s(\phi(s,a), \phi(s_j, a_j))}{h}\right) K'\left(\frac{d_r(w, w_j)}{h'}\right)}{\sum_{\ell=1}^m K'\left(\frac{d_r(w, w_l)}{h'}\right)},
\end{aligned}
\tag{8}
$$

where $d_r$ measures the similarity between weight vectors, and $d_s$ is the distance between state-action tuples projected into a lower dimension using $\phi$. The posterior is then

$$
\begin{aligned}
\widehat{p}_m^n(w|\{s_i^e, a_i^e\}_{i=1}^n) &\propto p(w) \prod_{i=1}^n \widehat{p}_m(s_i^e, a_i^e \mid w) \\
&= p(w) \prod_{i=1}^n \sum_{j=1}^m \frac{K\left(\frac{d_s(\phi(s_i^e, a_i^e), \phi(s_j, a_j))}{h}\right) K'\left(\frac{d_r(w, w_j)}{h'}\right)}{\sum_{\ell=1}^m K'\left(\frac{d_r(w, w_l)}{h'}\right)}.
\end{aligned}
\tag{9}
$$

The procedure for sampling from the posterior is identical, except that Equation 8 is used to calculate the likelihood, and Equation 9 to update the posterior.

## 5 Theoretical Guarantees

Before we investigate our method empirically, we prove asymptotic posterior consistency. Posterior consistency ensures that as the number of samples increases, the posterior distribution centers around the true reward function parameter value. This is important to ensure that our method is reliable at incorporating new data, and given enough data, will accurately learn the parameters corresponding to the true reward function.

KD-BIRL uses a probability density function to approximate the likelihood; consequently, we can reason about its posterior's asymptotic behavior. In particular, we show that the posterior estimate contracts as it receives more samples. This type of analysis also aligns with the behavior that we want in practice; more demonstration samples should shrink the posterior distribution around the true reward function.

We first focus on the likelihood estimation step, and we show that, when the size of the training dataset samples, $m$, approaches $\infty$ and the $m$ samples are associated with reward functions that sufficiently cover $\mathcal{R}$, the likelihood approximation (Equation 5) converges to the true likelihood $p(s, a \mid R)$.

**Lemma 5.1.** *Let $h_m, h'_m > 0$ be the bandwidths chosen to estimate the joint probability density and marginal probability density respectively. Assume that both the likelihood $p(s, a \mid R)$ and prior $p(R)$ are square-integrable and twice differentiable with a square-integrable and continuous second order derivative, and that $mh_m^{p/2} \to \infty$ and $mh'^{p'/2}_m \to \infty$ as $m \to \infty$, where $p$ is the dimension of $(s, a, R)$ and $p'$ is the dimension of $R$. Then,*

$$\widehat{p}_m(s, a|R) \xrightarrow[m \to \infty]{P} p(s, a \mid R), \ \forall (s, a, R) \in \mathcal{S} \times \mathcal{A} \times \mathcal{R},$$

*where $\widehat{p}_m(s, a|R)$ is the approximation of the likelihood using $m$ training dataset samples.*

Lemma 5.1 verifies that we can approximate the true likelihood function using CKDE, which opens the door to Bayesian inference. Because the IRL problem is non-identifiable, the "correct" reward function may not be unique (Abbeel & Ng, 2004; Ratliff et al., 2006b; Ziebart et al., 2009). In this work, we assume that any two reward functions that lead an agent to behave in the same way are considered equivalent. That is, if a set of demonstration trajectories is equally likely under two distinct reward functions, the functions are considered equal: $R_1 \simeq R_2$ if $\|p(\cdot|R_1) - p(\cdot|R_2)\|_{L_1} = 0$. Using this intuition, we can then define the *equivalence class* $[R^\star]$ for $R^\star$ as $[R^\star] = \{R \in \mathcal{R} : R \simeq R^\star\}$. Our objective is to learn a posterior distribution that places higher mass on reward functions that are in the equivalence class $[R^\star]$.

We now show that as $n$, the size of expert demonstration dataset for the test task, and $m$, the size of the training demonstration dataset, approach $\infty$, the posterior distribution contracts to the equivalence class of the reward function that best explains the expert demonstration dataset $[R^\star]$.

**Theorem 5.2.** *Assume the prior for $R$, denoted by $\mathcal{P}$, satisfies $\mathcal{P}(\{R : \mathrm{KL}(R^\star, R) < \epsilon\}) > 0$ for any $\epsilon > 0$, where $\mathrm{KL}$ is the Kullback–Leibler divergence. That is, prior $\mathcal{P}$ defines a non-empty set of possible reward functions such that each reward function is within a KL-divergence distance of $\epsilon$ from the data-generating reward function $R^\star$. Assume $\mathcal{R} \subseteq \mathbb{R}^d$ is a compact set. Let $\widehat{p}^n_m$ be the posterior density function estimated using $n$ test task samples and $m$ training task samples. Then, the posterior measure corresponding $\widehat{p}^n_m$ (Equation 6), denoted by $\mathcal{P}^n_m$, is consistent w.r.t. the $L_1$ distance; that is,*

$$\mathcal{P}^n_m(\{R : \|p(\cdot|R) - p(\cdot|R^\star)\|_{L_1} < \epsilon)\}) \xrightarrow[n \to \infty]{m \to \infty} 1.$$

Theorem 5.2 verifies that the posterior $\mathcal{P}^n_m$ assigns almost all mass to the neighborhood of $[R^\star]$. This means that KD-BIRL asymptotically contracts to the equivalence class of the test task data-generating reward function $R^\star$. Note that this not a statement regarding the posterior contraction rate, just a certification of contraction. Proofs for both Theorem 5.2 and Lemma 5.1 are in Appendix A.

## 6 Experiments

Building on our theoretical analysis, we seek to answer the following questions with our empirical analyses:

1. How does KD-BIRL perform in comparison to baseline methods (Section 6.1) under the standard reward function parameterization?

2. How does the featurized reward function parameterization improve performance? (Section 6.4)

3. How much quicker can KD-BIRL contract in comparison to baseline methods? (Section 6.4)

To answer these questions, we investigate two environments. The first is a *Gridworld* setting with a discrete state space. We use three grid sizes ($2 \times 2$, $5 \times 5$ and $10 \times 10$) to investigate how KD-BIRL's performance scales. The second setting is a simulated *sepsis treatment* environment (Amirhossein Kiani, 2019), which has a continuous state space and is thus, more challenging. The objective of the sepsis environment is to administer a series of treatment such that a patient is successfully discharged.

## 6.1 Baselines

We evaluate KD-BIRL's empirical performance in comparison to three Bayesian IRL baselines. We focus on Bayesian IRL approaches because the inferred posterior distribution provides uncertainty estimates of the reward function, which frequentist approaches fail to provide. Our first baseline is a single-task Bayesian IRL (BIRL) approach (Ramachandran & Amir, 2007), which we compare to in the environments with smaller state spaces. We avoid this comparison in the larger state-space environments because BIRL is computationally prohibitive to run. Next, we compare to a more recent single-task Bayesian IRL method, Approximate Variational Reward Imitation Learning (AVRIL) (Chan & van der Schaar, 2021). AVRIL uses variational inference to approximate the posterior distribution on the reward function, which is more computationally efficient at the cost of approximate inference. As stated earlier, one goal of our evaluations is to verify that incorporating data from training tasks improves the contraction rate of the posterior. Thus, we also compare to a baseline in which AVRIL is initialized using an informative prior learned from the training tasks. In this way, AVRIL can also take advantage of the information stored in the training task dataset. We initialize the mean and variance parameters of the AVRIL prior using the reward function samples from the training dataset. Specifically, given a training dataset, $\{(s_j, a_j, R_j)\}_{j=1}^m$, $p(R) = \mathcal{N}(\mu_0, \sigma_0^2)$, where $\mu_0 = \frac{1}{m} \sum_{j=1}^m R_j(s_j, a_j)$ and $\sigma_0^2 = \frac{1}{m} \sum_{j=1}^m (R_j(s_j, a_j) - \mu_0)^2$.

## 6.2 Evaluation Metrics

To evaluate the learned posterior distributions of the reward functions, we follow earlier work and use expected value difference (EVD) (Choi & Kim, 2012; 2013; Brown & Niekum, 2018; Levine et al., 2011). EVD measures the difference in total reward obtained by an agent whose policy is optimal for the true (data-generating) reward function and an agent whose policy is optimal for some estimated reward function. Define the value function of a policy $\pi$ under the reward function $R$ as,

$$V^{\pi,R}(s) = \mathbb{E}_\pi \left[ \sum_{t=0}^\infty \gamma^t R(S_t, A_t) \middle| S_0 = s \right].$$

Then, EVD is calculated as

$$EVD = |V^{\pi^\star, R^\star} - V^{\pi(r^L), R^\star}|,$$

where $\pi^\star$ and $\pi(r^L)$ are the optimal policies for the true reward function $R^\star$ and a reward function $r^L$ drawn from the posterior, respectively. EVD is a suitable metric for comparing the performance of methods without directly comparing the learned reward functions, which may have distinct functional forms. The lower the EVD, the better the estimated reward function $r^L$ captures the qualities of the true reward function $R^\star$ (details in Appendix H).

In addition to EVD, which evaluates the posterior concentration at the true reward function parameters, we plot the marginalized posterior distributions, which highlight how well KD-BIRL captures uncertainty.

## 6.3 Original reward function parameterization

In our first set of evaluations, we represent $R$ as a vector in which each cell contains a parameter corresponding to the scalar reward in one of the states. For example, in a $10 \times 10$ Gridworld environment, there are 100 distinct states, so the reward function $R \in \mathbb{R}^{100}$ is represented as a vector of length 100.

We first demonstrate that KD-BIRL's marginalized posterior distribution is more concentrated than baseline approaches in a $2 \times 2$ Gridworld. Here, the test task demonstrations arise from the reward function $R^\star = [0, 0, 0, 1]$. In addition to these test demonstrations, KD-BIRL receives training task demonstrations from the reward functions $R^1 = [1, 0, 0, 0]$ and $R^2 = [0, 1, 0, 0]$. To evaluate our method, we visualize the density of the posterior samples for each of the baselines, marginalized at each state.

The posterior samples from KD-BIRL are more concentrated around $R^\star$ than baseline approaches (Figure 1). The BIRL posterior samples are also close to the ground truth reward, but not as concentrated as those of KD-BIRL. In contrast, neither AVRIL nor AVRIL with an informative prior has concentrated posteriors in all of

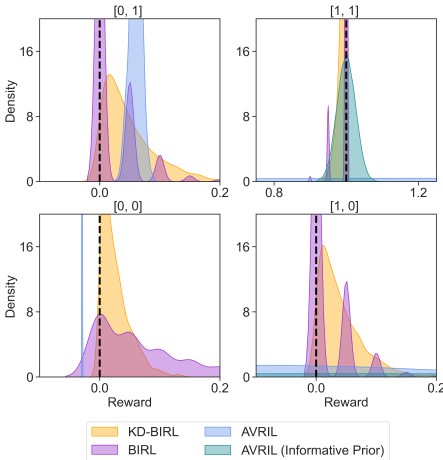

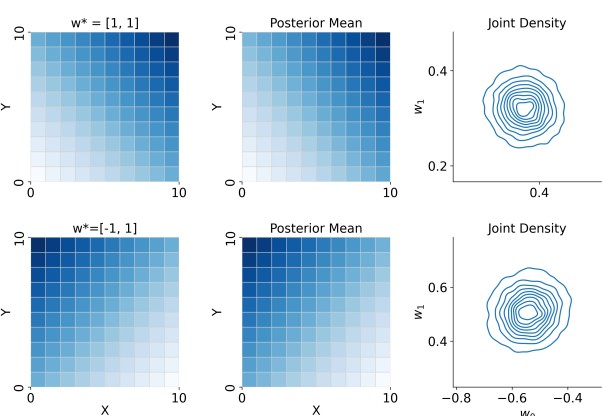

Figure 1: KD-BIRL learns more concentrated posteriors towards $R^\star$ than baseline methods. Dashed vertical lines display the true reward. The $x$ and $y$-axis represents the sampled reward marginalized at the given state and estimated density, respectively. A perfect marginalized posterior distribution would have the highest density at the black line ($R^\star$) in each state.

Figure 2: Using known features to parameterize reward enables reward inference in two settings in a 10x10 Gridworld. The first column shows $w^\star$ projected onto the Gridworld, the second shows the KD-BIRL posterior mean, and the third shows the joint density plots of the two weights, with the first on the x-axis and the second on the y-axis. KD-BIRL accurately infers the relative magnitude and sign of the individual weights.

the states. Notably, in state $[0, 1]$, these distributions are mostly uniform, when they should be concentrated at zero. This result indicates that using the combination of the training- and test task demonstrations can improve concentration of the posterior, but an informative prior is insufficient to make use of this information. In contrast, incorporating this information using the CKDE-based likelihood formulation can lead to concentrated state-marginalized posterior distributions.

While the $2 \times 2$ Gridworld experiment is promising, we would like to apply our method in environments with much higher-dimensional reward functions. As discussed in Section 4, kernel density estimation scales poorly as the number of reward function parameters increase (Izbicki & Lee, 2017). To support this claim and motivate the use of a feature-based reward function, we show results under the original reward function parameterization in a $5 \times 5$ Gridworld environment.

We select a data-generating reward function $R^\star$ that has a nonzero reward in only one of the states (Figure 3, Panel 1). The state occupancy of the test task expert demonstrations can be seen in Figure 3, Panel 4. From Panel 4, we see that the test task expert demonstrations all approach the upper right corner of the grid. KD-BIRL also receives training task demonstrations from three other reward functions. The state occupancy density of the training demonstrations can be seen in Figure 3, Panel 3. In contrast with the test task expert demonstrations, the training task demonstrations cover far more states in the environment. Thus, the training task demonstrations contain information about portions of the state-space that the expert demonstration do not cover, which should ideally improve posterior distribution inference.

In our results, we find that our method is able to estimate a posterior whose mean is in the equivalence class of $R^\star$ (Figure 3, Panel 2). In other words, an agent that follows the reward function identified by the posterior mean would still be directed to the terminal state in the upper right corner of the grid (which contains the highest reward). However, despite having demonstrations from four separate reward functions across the training and test tasks, the posterior mean is slightly inaccurate. There are states ($[2, 3], [4, 3]$ in Figure 3, Panel 2) where the estimated scalar reward is incorrect, which suggests that the CKDE struggles to learn 25 independent reward parameters successfully from these diverse demonstrations. This result motivates the

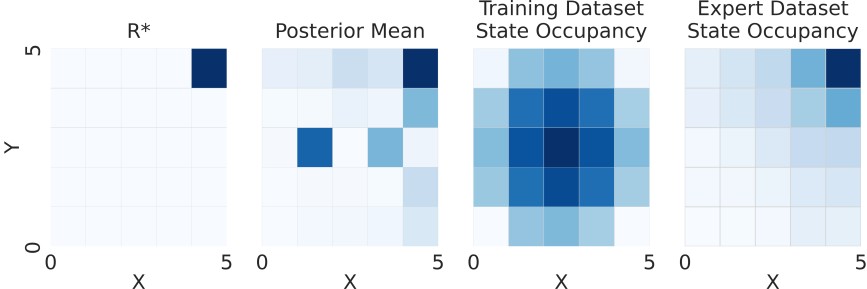

Figure 3: We study KD-BIRL's performance in a $5 \times 5$ Gridworld environment in which the reward is positive in one state (upper right corner). The first panel shows the data generating reward function $R^\star$, the second panel shows the KD-BIRL posterior mean, and the third and fourth panels show the state occupancy distribution of the training task and expert task demonstrations respectively. The KD-BIRL posterior mean is within the equivalence class of $R^\star$, because the darkest square is in the upper right hand corner, but there are incorrect estimates of reward in other states.

## 6.4 Feature-based reward function

Now, we investigate a feature-based reward function, which enables KD-BIRL to perform posterior inference in environments with a high-dimensional state space. To build intuition for this approach, we first investigate a $10 \times 10$ Gridworld environment, where the state space $\mathcal{S}$ is the series of vectors of length 100. Here, we select $\phi(s, a) = [x, y]$ to be a function that maps the state vector to the spatial coordinates of the agent. That is, we treat the coordinates of the agent as a known "feature vector". In this setting, the reward function is $R^\star = w^{\star T} \phi(s, a)$ where $w^\star$ is a low-dimensional weight vector that we aim to learn. Similar to prior domains, KD-BIRL receives training task samples from two other reward functions each parameterized by corresponding weight vectors.

Our results indicate that a feature-based reward function allows KD-BIRL to perform reward inference in a $10 \times 10$ Gridworld under two different reward functions. First, we note that the posterior mean, when projected onto the Gridworld, is nearly identical to the ground truth weight vector $w^\star$ projected onto the Gridworld. This indicates that the learned posterior is concentrated close to $w^\star$, which is ideal. Next, we visualize the learned posterior parameters for both weights, and find that KD-BIRL is able to accurately recover the relative magnitude and sign of the individual weights $w$ in two settings with different $w^\star$ values (Figure 2). Our results in the $10 \times 10$ Gridworld setting demonstrate that KD-BIRL is able to infer reward functions in environments with high-dimensional ($R \in \mathbb{R}^{100}$) state spaces using a featurized reward function.

Next, we investigate KD-BIRL's performance in the sepsis treatment environment, which presents additional challenges due to its continuous state space. Unlike the previous Gridworld environments, each state in the sepsis environment is a vector of length 46, where each element corresponds to a real-valued measurement of a given feature. As a result, the reward function parameterization needs to learn a low-dimensional representation of the state features. To do this, we learn $\phi$ using a variational auto-encoder (VAE). The use of a VAE is appropriate in this situation because we do not know in advance which features will be most relevant to reward function inference. The VAE takes as input vectors of length 47 (46 state features + 1 treatment action), and projects them into a latent space that has 3 features.

Now, we seek to understand how KD-BIRL performs when it receives few test task expert demonstrations in comparison to the size of the training task dataset. KD-BIRL receives expert demonstrations from one unknown reward function, which is parameterized using a set of weights $w^\star$, and training task demonstrations from four additional known reward functions. Our baselines include AVRIL, which only receives the expert

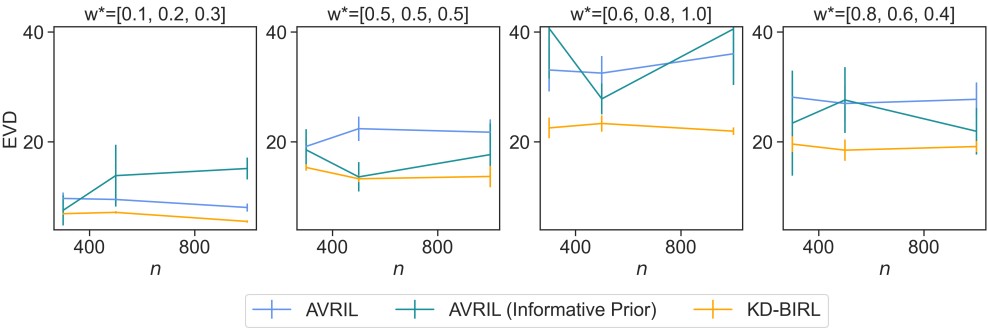

Figure 4: **The KD-BIRL posterior empirically contracts faster than baselines**. Here, we evaluate four sepsis treatment settings, each with a distinct $w^\star$. The $x-, y-$ axes represent the number of test demonstrations and the EVD of the learned posterior respectively, and the error bars indicate standard error. We compare to two baselines, AVRIL and AVRIL with an informative prior. Our results indicate that KD-BIRL is able to produce lower or comparable EVD with fewer samples than either baseline in a low expert demonstration data regime ($n < 1000$).

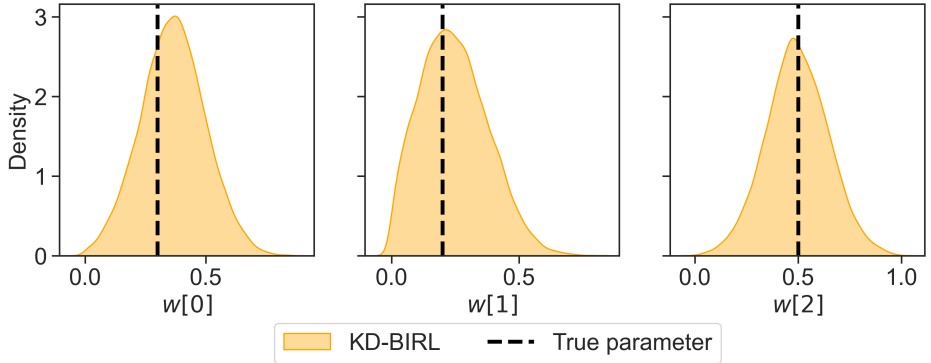

Figure 5: The marginal posterior of KD-BIRL for a chosen $w^\star$ in the sepsis environment: The true parameter is displayed in the black dashed line, and the density of the posterior marginalized at each parameter is plotted on three plots.

demonstrations, and AVRIL with an informative prior, which uses a prior that is calculated based on the mean and variance of the demonstrations in the training task dataset.

In four settings, each with a different $w^\star$, KD-BIRL's posterior samples generate consistently lower EVD than AVRIL (Figure 4), and notably lower standard error than AVRIL with an informative prior. In particular, KD-BIRL achieves lower EVD in the low-expert demonstration data regime where there are fewer than 1000 samples in the test expert demonstration dataset. Lower EVD implies that the posterior is more concentrated at $w^\star$. Figure 4 shows that as $n$ increases, the EVD of KD-BIRL is lower than both baselines, which implies that the KD-BIRL posterior is more concentrated with the same number of samples. This indicates that KD-BIRL can learn a more accurate posterior with fewer test demonstration samples by leveraging information from the training tasks. Taken together, this suggests that the KD-BIRL posterior requires fewer samples to contract to $w^\star$. We also note that the informative prior that is learned using the training task demonstrations has a large variance, which we believe leads to posterior samples that produce EVDs with high variance.

Finally, as noted earlier, one of the benefits of using a Bayesian method is uncertainty quantification. We visualize the learned posterior parameters for a fifth setting with a new $w^\star$ (Figure 5), and find that the posterior is concentrated at the true parameter. Our experiments demonstrate that using a VAE to learn

$\phi$ can effectively summarize state information in continuous and high-dimensional state spaces. This result, coupled with asymptotic posterior consistency guarantees, makes KD-BIRL a superior choice for performing IRL when there are few test task expert demonstrations available.

## 7 Discussion

In this work, we present kernel density Bayesian inverse reinforcement learning (KD-BIRL), an IRL algorithm that uses CKDE to incorporate existing domain-specific datasets to improve the posterior contraction rate. We address a gap in the Bayesian IRL literature by leveraging a consistent likelihood estimator and translating this estimator into posterior consistency guarantees for this class of methods. Our experiments demonstrate that KD-BIRL can perform IRL in high-dimensional and continuous state spaces. Notably, KD-BIRL learns a more concentrated posterior distribution with fewer samples, outperforming a leading single-task IRL method, even when this leading method is initialized with an informative prior. Our results show that KD-BIRL can learn posterior distributions that capture key characteristics of objectives while providing uncertainty estimates, even with limited data.

**Limitations** Our work assumes that the expert demonstrations are of high quality; we anticipate that training task demonstrations that are not optimal w.r.t the corresponding reward function will result in less accurate likelihood estimates. Furthermore, we assume that the features used in the feature-based reward function fully capture the information stored within a state-action tuple. Further work is needed to analyze the effect of less representative features. Our work also assumes that the L1 norm is a valid distance metric between posterior densities, and this may not be true depending on the application setting.

**Future Work** Several future directions remain open. The particular choices of distance metrics and hyperparameters used in the CKDE depend on the environment and reward function parameterization; additional experimentation is required to characterize the relationship between these metrics and the quality of the learned posterior. Furthermore, it is of interest to explore other ways in which the CKDE can be used to infer higher dimensional reward functions. This may include exploring how to speed up the CKDE (Holmes et al., 2007), or replacing it with another nonparametric conditional density estimator (Rojas et al., 2005; Hinder et al., 2021). Additionally, our work proposes using density estimation to approximate the likelihood, and there are several other choices for learning this density function including diffusion models (Cai & Lee, 2023), and Gaussian processes (D'Emilio et al., 2021).

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
