# OpenReview forum: "KD-BIRL: Kernel Density Bayesian Inverse Reinforcement Learning"
_TMLR — Accepted by TMLR_

### Review · Reviewer_SgoL · 2024-08-22

**Summary Of Contributions:**

The paper introduces a new algorithm to solve Inverse Reinforcement Learning tasks in the setting where we have demonstrations as well as full knowledge of the reward functions for some training tasks and only a few expert demonstrations for the target task. Within the Bayesian framework, it uses a Conditional Kernel Density approach to estimate the likelihood function and integrate information from the training tasks. The paper also proves asymptotic consistency of the approach used. Overall the methods presented are novel and the results are promising, especially in an environment where the state-action space is of high dimension.

**Audience:**

Yes

**Claims And Evidence:**

Yes

**Requested Changes:**

These proposed adjustments would strengthen the work:
- Add further study of the application of the method in real-world and non-generated settings - see Weaknesses section above
- Allocate more time studying rate of contraction and uncertainty reduction as a function of the number of demonstration
- Add more explanation about each step of the application of KD-BIRL. Some parts are particularly dense and could benefit from more clarification and formalization (e.g. sections 6.3 and 6.4)
- Typo in p8: should be $\sigma_0^2$ instead of $\sigma_0$

**Strengths And Weaknesses:**

Strengths:
- The Conditional Kernel Density Estimation (CKDE) approach is a well-conceived and reusable framework. It incorporates information from alternative demonstrations, allowing for a more efficient convergence to the true reward function of the target task compared to other known methods.
- The paper is clearly written, and the experimental results are robust, particularly in high-dimensional settings
- The workflow presented for addressing high-dimensional IRL problems is both effective and generalizable. The use of a low-dimensional representation of the reward functions such VAEs employed here, combined with CKDE, provides a practical and reusable strategy for similar tasks.
- The theoretical consistency guarantees provided by the authors offer a strong mathematical foundation for the approach

Weaknesses:
I believe the main weakness of the paper is the application of this novel method in only simple if not toy settings. The majority of the experiments discussed are variations of Gridworld, while the sepsis setting is still treated through generated reward functions. Given the potential of the method, it would be good to see how it behaves in a non-generated setting.

---

> ### Author Response · Authors · 2024-09-04
> **Author Response**
>
> **Additional domains**: As mentioned in the overall comment, the goal of our experiments section is to assess the performance of our algorithm in settings that reflect real-world application scenarios. In particular, real-world application scenarios have high-dimensional state spaces, and are continuous environments. In our work, we test our algorithm in both discrete (Gridworld), and continuous (Sepsis) environments. We also test the ability of our approach in low-dimensional environments (2x2 Gridworld, 4 states), and high-dimensional environments (10x10 Gridworld=100 states and Sepsis environment=continuous state space with 54 dimensions). More particularly, the Sepsis environment is realistic, complicated, and reflects real-world medical settings well. We believe that our evaluations are sufficient to ensure the applicability of our approach, and further work will involve applying our method to real-world settings. We believe that our work is a precursor to deploying in a “real-world” setting, and that the evaluations in simulated environments are sufficient for the scope of the project.
>
> **Contraction rate**: In our work, we approximate the unknown likelihood function using kernel density estimation (KDE). In this setting, proving the contraction rate is challenging because the likelihood is approximated, rather than fully known. The approximation error varies with the size of the demonstration datasets and may not be uniform across the parameter space, making it difficult to apply standard techniques for establishing contraction rates. Recall that in Bayesian asymptotics, contraction rates are typically derived under assumptions that the likelihood function is fully known and that it satisfies certain regularity conditions. While our current focus is posterior consistency, we acknowledge that the contraction rate is an interesting future direction, which requires developing new methods or extending existing Bayesian asymptotic theories to handle the complexities introduced by the likelihood approximation. In particular, we may look into ideas from literature on Bayesian asymptotics with misspecified models, such as the work in [1]. Adapting and extending their ideas to our context, where the likelihood is approximated rather than fixed to be misspecified, could lead to a potential solution to the contraction rate.
>
> **Additional explanation and typos**: We have clarified our algorithm setup in section 6.3 and 6.4. In particular, we expanded our explanation of the featurization of our observations, and have clearly outlined the conclusions that are meant to be drawn from each section of our experiments. We have also fixed the typos noted.
>
> [1] De Blasi, P., & Walker, S. G. (2013). BAYESIAN ASYMPTOTICS WITH MISSPECIFIED MODELS. Statistica Sinica, 23(1), 169–187. http://www.jstor.org/stable/24310519

---

### Review · Reviewer_bBST · 2024-08-25

**Summary Of Contributions:**

The paper introduces a new method called Kernel Density Bayesian Inverse Reinforcement Learning (KD-BIRL). This method enhances existing Bayesian IRL approaches, particularly in situations where expert demonstration data is scarce. The key idea is to use conditional kernel density estimation (CKDE) to incorporate additional data from training tasks, which improves the accuracy of the learned reward function. The authors also provide theoretical guarantees on the method’s ability to concentrate on the correct reward function as more data becomes available. Empirical results show that KD-BIRL outperforms traditional approaches in both discrete (Gridworld) and continuous (sepsis treatment simulation) environments.

**Audience:**

Yes

**Broader Impact Concerns:**

N/A.

**Claims And Evidence:**

Yes

**Requested Changes:**

See weaknesses for the details.

**Strengths And Weaknesses:**

I want to preface this section by stating that I am not an expert in this area so I apologize for any inaccuracies.

## Strengths:
- The integration of CKDE into Bayesian IRL allows the method to perform better in low-data scenarios by leveraging additional information from training tasks.
- The paper provides a solid theoretical basis for the method, though it relies on existing theoretical results, making the analysis relatively straightforward.
- The experiments demonstrate the method’s effectiveness in both discrete and continuous environments.

## Weaknesses:
- The computational demands of kernel density estimation might limit the method’s applicability to high-dimensional problems.
- The use of the \(L_1\) norm in Theorem 5.2 and the assumption \(P(\{R: KL(R^\ast, R) < \epsilon\}) > 0\) could be more clearly explained. The assumption might be better expressed as stating that there exists a non-empty set satisfying the condition.
- The paper does not explicitly discuss the method’s limitations, particularly with respect to scalability and assumptions made in the theoretical analysis.
- I think the authors could test their KD-BIRL algorithms on more environments to further demonstrate its generalizability and robustness.

---

> ### Author Response · Authors · 2024-09-04
> **Author Response**
>
> **Assumption about the L1 norm**: Intuitively, this assumption means that the prior places positive mass to any neighbor of the true parameter $R^\star$. Otherwise, if the prior placed zero mass on this neighbor of the true parameter, then the posterior would also place zero mass on the same neighborhood, regardless of the observation data. This posterior would therefore not be consistent. Such an assumption on the prior distribution is standard in Bayesian theory- one example of a similar assumption is in Section 1.3 of Bayesian Nonparametrics [1].
>
> **Computational demands of KDE**: KDE does increase in computational complexity with higher dimensional settings. Our work notes this and uses a feature-based reward function to reduce the dimension of the reward function in higher dimensional settings. In particular, this feature-based approach can enable our method to perform reward function inference in high-dimensional settings such as the Sepsis environment, which has a continuous state space with 54 dimensions.
>
> **Limitations**: We have included a more thorough discussion about the limitations of our approach in the conclusion. These limitations include the fact that we assume the L1 norm is a good way to measure the difference between reward functions, and this may not be true. Additionally, the KDE is slow in higher dimensional environments; while we propose a feature-based reward function to address this, if we do not have a low-dimensional featurization, or the featurization does not capture the reward function well, our method would not scale well.
>
> **Further environments**: As mentioned in the overall comment, the goal of our experiments section is to assess the performance of our algorithm in settings that reflect real-world application scenarios. In particular, real-world application scenarios have high-dimensional state spaces, and are continuous environments. In our work, we test our algorithm in both discrete (Gridworld), and continuous (Sepsis) environments. We also test the ability of our approach in low-dimensional environments (2x2 Gridworld, 4 states), and high-dimensional environments (10x10 Gridworld=100 states and Sepsis environment=continuous state space with 54 dimensions). More particularly, the Sepsis environment is realistic, complex, and reflects real-world medical settings well. We believe that our evaluations are sufficient to ensure the applicability of our approach, and further work will involve applying our method to real-world settings.
>
> [1] Ghosh, J. & Ramamoorthi, R. (2011). Bayesian Nonparametrics. Springer Series in Statistics. 16.

---

### Review · Reviewer_SESx · 2024-08-27

**Summary Of Contributions:**

This work proposes KD-BIRL, a bayesian inverse reinforcement learning method that leverages a training set of expert demonstrations and reward functions in order to achieve better performance on estimating a new reward function given a new set of expert demonstrations. The method is based on a kernel density estimator for approximating the likelihood of the observations given a reward function. The method is theoretically shown to converge to the correct equivalence class of reward functions and empirically shown to have a better performance than the baselines.

**Audience:**

Yes

**Broader Impact Concerns:**

It could be important to refer the implications of the inclusion of non-expert demonstrations and rewards on the training sets, as well as the impact of the choice of features.

**Claims And Evidence:**

Yes

**Requested Changes:**

My minor recommendation is that "posterior contraction" and "posterior concentration" are either explained or replaced by common-knowledge terms early on in the paper (including the abstract and introduction).

More relevant, I would also like to see clarified the third bullet I mentioned above in the empirical analysis section (where and how was the contraction rate analyzed and against which baselines).

Lastly, I think it would be important to see mentioned/discussed the impact of non-expert demonstrations and rewards on the training sets as well as the impact of the choice of features.

**Strengths And Weaknesses:**

The problem of inverse reinforcement learning is much relevant in both theory and practice and this work presents a significant step forward in the field. The paper is clear and sound, and the method is sufficiently novel.

One weakness I point out is minor and is the use of jargon from very early on in the paper. Terms such as "posterior contraction", "posterior concentration", and perhaps even "posterior", "prior" and "likelihood" may not be perfectly clear to the general reader.

My other question is regarding the empirical analysis. The author say their third question empirical question is whether their method contracts quicker than baselines ("How much quicker can KD-BIRL contract in comparison to baseline methods? (Section 6.4)", bullet 3 in the first paragraph of Section  6). I do not understand which results/figures correspond to that specific question. In other words, I don't see/understand how the contraction rate was analyzed and against which baselines and would appreciate the authors clarifying.

Finally, I also think it would have been important to see discussed how the quality of the expert demonstrations may affect performance, as well as the choice of features.

---

> ### Author Response · Authors · 2024-09-04
> **Author Response**
>
> **Teminology**: We have modified our paper to clarify the terms suggested by the reviewer as well as a few other terms. For reference, here are definitions that we added to our paper:
> - Posterior contraction: A Bayesian posterior changes depending on the samples that are observed. Contraction is the phenomenon of concentrating at the true value as more samples are observed.
> - Posterior distribution: This is the parameterized distribution over reward functions after observing all samples.
> - Prior distribution: This is the parameterized distribution before observing samples. Prior distributions can be uninformative (i.e., uniform), or informative (i.e., Gaussian).
> - Likelihood: The probability distribution that defines how likely a sample is to be observed, given parameter values for a specific probabilistic model.
>
> **Empirical Analysis**: Our empirical analysis of the concentration rate is summarized in Figure 4. The goal here is to understand how many samples are required for the learned posterior to have sufficiently low Expected Value Difference (EVD). In this figure, we compare KD-BIRL to two baselines, which are AVRIL and AVRIL with an informative prior, and find that KD-BIRL is able to concentrate with fewer samples than either baseline. We have clarified this discussion in Section 6.4.
>
> **Effect of the quality of expert demonstrations**: Our work, like many inverse RL algorithms, assume that our samples are indeed optimal with respect to some unknown reward function. However, if this assumption is violated, we expect our posterior distribution to be less accurate. We have clarified this as an assumption of our work, and future work will study the effect of the quality of the expert demonstrations on the learned posterior distribution.
>
> **Effect of feature choice**: The feature choice for the state-action representation does indeed affect the quality of the learned posterior. In our approach, we use features that fully capture the information stored in the state-action tuple. For example, in the 10x10 Gridworld, we use the (x,y) coordinates, which fully capture the information stored in the state; in the Sepsis environment, we use a VAE, which projects each state into a low-dimensional representation and minimizes representation error. We hypothesize that if these featurizations were worse, the posterior would not contract as well. We have added a discussion about this point in the featurization section and conclusion.

---

### Author Response · Authors · 2024-09-04
**Overall comment**

We thank the reviewers for their detailed comments. We are encouraged that the reviewers found that our work “presents a significant step forward in the field” (SESx) and that our algorithm has a “strong mathematical foundation” (SgoL). One main concern from reviewers is the generalizability of our approach to additional domains. In our work, we first used a low-dimensional environment (2x2 Gridworld, 4 states) to validate the performance of our algorithm. We further designed domains to reflect real-world application scenarios, which often have high-dimensional and continuous state spaces. In particular, we tested our approach on several high-dimensional environments (10x10 Gridworld, 100 states) and a Sepsis environment (continuous state space with 54 dimensions). The Sepsis environment is fairly complicated and is designed to reflect real-world medical settings. We appreciate the reviewers’ feedback and agree that our work may benefit from testing on additional real-world applications. We will definitely pursue this as a future direction. We address reviewer-specific comments in individual responses, and have revised the manuscript to reflect these comments.

---

### Author Response · Authors · 2024-10-23
**Minor Revisions**

We thank the action editor and reviewers for their feedback and suggestions. We have uploaded a revised version of our manuscript that addresses each of these suggestions. We note the following changes in our manuscript in order of appearance:
1. Terminology: We have introduced the definitions for terminology used in the paper in Section 1.
2. Effect of expert demonstration quality: We clarify that we assume optimal expert demonstrations in Section 3.2, and discuss the impact of sub-optimal demonstrations in Section 7.
3. Equations 3 and 4: We have made a note that these equations do not define proper distributions in Section 3.3.
4. Effect of feature choice: We note in Section 4.2 that our method assumes that the set of features fully captures the information stored in the state-action tuple. We also mention this as a limitation in Section 7.
5. Assumption for theory: We expand the discussion for the assumptions in Theorem 5.2 (Section 5). We also note that the use of the L1 norm may not be appropriate in Section 7.
6. Empirical Analysis of posterior contraction rate: We expand the discussion of posterior contraction rate in Section 6.4.
7. Limitations: We include a discussion of limitations in Section 7.
8. Explanation about each step in the KD-BIRL algorithm: We add an algorithm box to Section G of the appendix to clarify our approach.

In addition to the above changes, we have fixed any mentioned typos and reframed our work to emphasize realistic applications less.

---

### Decision · Action_Editor_tuVm · 2024-09-21

**Recommendation:** Accept with minor revision

**Comment:**

This paper proposes a kernel-based method for Bayesian inverse reinforcement learning called KD-BIRL. The key idea in the method is to reuse pairs of reward functions and demonstrations from similar tasks to estimate the likelihood of demonstrations in a new task with an unknown reward function. This in turn allows computing the posterior distribution of the reward function in the new task. KD-BIRL is both analyzed and empirically evaluated.

Most comments of the reviewers were minor (clarify terminology, clarify notation, state limitations) and the authors addressed them in the revised manuscript. However, I checked and the revised manuscript does not seem to be submitted. Therefore, I accept this paper with a minor revision, and will make the final decision after I see the changes.

Two more things:

* One comment of the reviewers was that the solved problems are not large scale and realistic by today's standards. I share this sentiment. However, I also believe that the evaluation is sufficient to support the claims of the paper. To address this, I suggest that the authors downplay the large-scale and realistic claims.

* I also looked at the paper. When you introduce kernel-based approximations in (3) and (4), make clear that these are not proper distributions or densities because they have not been normalized.

**Audience:**

The audience for this paper are RL practitioners. The proposed method is easy to implement and thus others can use it as a baseline.

**Claims And Evidence:**

The proposed method KD-BIRL is both analyzed and empirically evaluated. Since KD-BIRL is simple and practical, I share the opinion of the reviewers that the empirical evaluation could have been done on larger and more realistic problems. That being said, the evaluation is sufficient to support the claims of the paper.

---

> ### Author Response · Authors · 2024-10-23
> **Is it appropriate to upload a camera ready revision?**
>
> We thank the action editor for the recommendations. I have uploaded an anonymized version of the paper with the requested modifications. Should I upload a camera ready revision that is de-anonymized?